

# Global sensitivity and uncertainty analysis of an atmospheric chemistry transport model: the FRAME model (v. 9.15.0) as a case study

Ksenia Aleksankina,[1,2] Mathew R. Heal,[1] Anthony J. Dore,[2] Marcel Van Oijen,[2] Stefan Reis[2,3]

[1] School of Chemistry, University of Edinburgh, Edinburgh, UK
[2] NERC Centre for Ecology & Hydrology, Penicuik, UK
[3] University of Exeter Medical School, European Centre for Environment and Health, Knowledge Spa, Truro, UK
Correspondence to: Mathew Heal (m.heal@ed.ac.uk)

**Abstract.** Atmospheric chemistry transport models (ACTMs) are widely used to underpin policy decisions associated with the impact of potential changes in emissions on future pollutant concentrations and deposition. It is therefore essential to have a
quantitative understanding of the uncertainty in model output arising from uncertainties in the input pollutant emissions. ACTMs incorporate complex and non-linear descriptions of chemical and physical processes which means that interactions and non-linearities in input–output relationships may not be revealed through the local one-at-a-time sensitivity analysis typically used. The aim of this work is to demonstrate a global sensitivity and uncertainty analysis approach for an ACTM, using as an example the FRAME model, which is extensively employed in the UK to generate source-receptor matrices for
the UK Integrated Assessment Model and to estimate critical load exceedances. An optimised Latin hypercube sampling design was used to construct model runs within ± 40 % variation range for the UK emissions of $SO_2$, $NO_x$ and $NH_3$, from which regression coefficients for each input-output combination and each model grid (>10,000 across the UK) were calculated. Surface concentrations of $SO_2$, $NO_x$ and $NH_3$ (and of deposition of S and N) were found to be predominantly sensitive to the emissions of the respective pollutant, while sensitivities of secondary species such as $HNO_3$ and particulate $SO_4^{2-}$, $NO_3^-$ and
$NH_4^+$ to pollutant emissions were more complex and geographically variable. The uncertainties in model output variables were propagated from the uncertainty ranges reported by the UK National Atmospheric Emissions Inventory for the emissions of $SO_2$, $NO_x$ and $NH_3$ (± 4 %, ± 10 % and ± 20 % respectively). The uncertainties in the surface concentrations of $NH_3$ and $NO_x$ and the depositions of $NH_x$ and $NO_y$ were dominated by the uncertainties in emissions of $NH_3$, and $NO_x$ respectively, whilst concentrations of $SO_2$ and deposition of $SO_y$ were affected by the uncertainties in both $SO_2$ and $NH_3$ emissions. Likewise, the
relative uncertainties in the modelled surface concentrations of each of the secondary pollutant variables ($NH_4^+$, $NO_3^-$, $SO_4^{2-}$ and $HNO_3$) were due to uncertainties in at least two input variables. In all cases the spatial distribution of relative uncertainty was found to be geographically heterogeneous. The global methods used here can be applied to conduct sensitivity and uncertainty analyses of other ACTMs.



# 1 Introduction

Atmospheric chemistry transport models (ACTMs) provide scientific support for policy development. It is therefore important to have a quantitative understanding of the levels of uncertainty associated with model outputs (AQEG, 2015; Frost et al.,
2013; Rypdal and Winiwarter, 2001). Sensitivity and uncertainty analyses are both used in this regard. Uncertainty analysis is applied to quantify propagation of uncertainties of single or multiple inputs through to a model output, whilst sensitivity analysis is used to investigate input-output relationships and to apportion the variation in model output to the different inputs. However, due to the complexity of ACTMs the relationship between model inputs and outputs is not analytically tractable so both quantities must be estimated by sampling model inputs according to an experimental design and undertaking multiple
model simulations (Dean et al., 2015; Norton, 2015; Saltelli et al., 2000; Saltelli and Annoni, 2010).

Typically, model assessment studies focus on uncertainties in the model parameter values (Derwent, 1987; Konda et al., 2010; De Simone et al., 2014) and model-specific structure (Simpson et al., 2003; Thompson and Selin, 2012). However, for ACTMs the uncertainty in the model input emissions data could be dominating; for example, previous dispersion model uncertainty studies identified input emissions as a primary source of uncertainty in model outputs (Bergin et al., 1999; Hanna et al., 2007;
Sax and Isakov, 2003). It is also the case that a major role of ACTMs is to estimate the impact of potential future changes in emissions on atmospheric composition (Boldo et al., 2011; Crippa et al., 2016; Heal et al., 2013; Vieno et al., 2016; Xing et al., 2011; Zhang et al., 2010).

Thus the focus of this study is to demonstrate a systematic approach for quantifying model output sensitivity and uncertainty as a function of the variation in model input emissions. We used the Fine Resolution Atmospheric Multi-pollutant Exchange
(FRAME) model as a case study. FRAME is a Lagrangian model that outputs, at a 5 km × 5 km horizontal resolution over the UK, annual average surface concentrations of sulphur dioxide ($SO_2$), nitrogen oxides ($NO_x$), ammonia ($NH_3$), nitric acid ($HNO_3$), and particulate ammonium ($NH_4^+$), sulphate ($SO_4^{2-}$), and nitrate ($NO_3^-$), together with dry and wet deposition of oxidised sulphur ($SO_y$), oxidised nitrogen ($NO_y$), and reduced nitrogen ($NH_x$) (Dore et al., 2012; Matejko et al., 2009; Singles et al., 1998). The model is extensively used to provide policy support including generation of source-receptor matrices for the
UK Integrated Assessment Model (UKIAM) and estimation of critical load exceedances (Matejko et al., 2009; Oxley et al., 2013). Source receptor matrices link concentration and deposition with individual emission sources and are used to automate procedures to estimate the impact of future emission reduction scenarios. Integrated assessment modelling incorporates technical emissions abatement costs with cost-benefit analysis and source-receptor data to indicate cost-effective solutions to protect natural ecosystems from acidic and nitrogen deposition above defined critical thresholds and to protect human health
from particulate concentrations (Oxley et al., 2003, 2013).

FRAME uses emissions input data from the UK National Atmospheric Emissions Inventory (NAEI, http://naei.beis.gov.uk/), which are compiled following the international 'Guidelines for Reporting Emissions and Projections Data under the

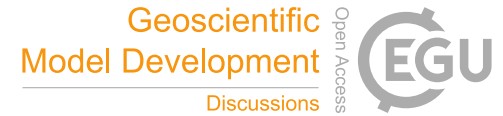

Convention on Long-range Transboundary Air Pollution (United Nations Economic Commission for Europe, 2015). We used the uncertainties published by the NAEI in the Informative Inventory Report (Misra et al., 2015) as the foundation of the uncertainty propagation for the FRAME concentration and deposition outputs with respect to UK emissions of $SO_2$, $NO_x$, and $NH_3$. The uncertainty ranges for different pollutants reported by the NAEI are estimated using a Monte Carlo technique which

corresponds to the IPCC Tier 2 approach (IPCC, 2006). In this approach uncertainty ranges for each source for both emission factor and activity statistics are associated with a probability distribution and further used as inputs in a stochastic simulation which calculates output distributions of total UK emissions for each pollutant. The uncertainties are expressed as plus or minus half the confidence interval width relative to the estimated emissions value.

Previously, local one-at-a-time (OAT) sensitivity analysis has been used to investigate ACTM sensitivity because it is less

computationally demanding than global sensitivity analysis that requires a large number of simultaneous perturbations of all inputs of interest. However, there are significant disadvantages associated with OAT analysis: the interactions between the input parameters and non-linearities in the model response cannot be identified; additionally as the number of input parameters increases the fraction of parameter space investigated tends to zero (Jimenez and Landgrebe, 1998; Saltelli and Annoni, 2010). Therefore local OAT sensitivity analysis is only applicable when the effects of the different inputs are all independent from

each other and model response is linear for the range of investigated inputs. Many previous publications that include ACTM sensitivity analysis use the OAT approach but fail to acknowledge its limitations (Appel et al., 2007; Borge et al., 2008; Capaldo and Pandis, 1997; Labrador et al., 2005; Makar et al., 2009).

Hence this study focuses on demonstrating the use of global methods capable of revealing non-linearity in the model response and the presence of interactions between inputs in addition to revealing the spatial pattern of the model response to changes in

the input emissions. Increasing computational resource means this approach is now starting to be applied to ACTMs (Christian et al., 2017).

In a global sensitivity analysis a sample space is created for all inputs under investigation from which a set of combinations of model inputs for different model runs are chosen. The sampling design for model inputs for uncertainty and sensitivity analysis must balance the needs of covering the full multidimensional input parameter space at sufficient density to allow

characterisation of any non-linearities and interactions in the model response with a small enough number of samples for the total number of model runs to remain computationally tractable. Simple random sampling is conceptually the simplest sampling technique, but has low efficiency compared to other sampling approaches and tends to lead to clusters and gaps in coverage of the input space (Saltelli et al., 2008). Likewise, full or fractional factorial designs (Box and Hunter, 1961) do not allow effective exploration of the whole input space because for more than a few levels of each input the number of model

runs becomes very large. Quasi-random sampling, of which the Sobol' sequence (Sobol', 1967, 1976; Sobol' and Levitan, 1999) is a popular choice for variance-based sensitivity analysis, may not work well when the number of sampling points is small (Saltelli et al., 2008). Therefore, in this work, Latin hypercube sampling (LHS) (McKay et al., 1979), which is a stratified space-filling sampling technique, was used. Advances have been made to optimise the space filling properties of LHS including



maximin sampling (Johnson et al., 1990; Morris and Mitchell, 1995) and integrated mean squared-error minimisation (Park, 1994).

In summary, this work demonstrates application of global uncertainty and sensitivity analysis to an ACTM using the FRAME model as an example.

## 5   2 Methods

### 2.1 Model description

The FRAME model is a Lagrangian model that calculates annual average surface concentrations of $SO_2$, $NO_x$, $NH_3$, and $HNO_3$, particulate $NH_4^+$, $SO_4^{2-}$, and $NO_3^-$, and dry and wet deposition of $SO_y$, $NO_y$ and $NH_x$ at 5 km × 5 km horizontal resolution over the UK (Dore et al., 2012; Fournier et al., 2002; Matejko et al., 2009; Singles et al., 1998). This spatial resolution corresponds

to >10,000 model grid squares over the UK land area. The air column contains 33 vertical layers of varying thickness from 1 m at the surface to 100 m at the top of the mixing layer. The vertical diffusion between layers is calculated using *K*-theory. The air columns move from the boundary of the domain along straight-line trajectories with varying starting angles at a 1° resolution. The trajectories are defined by an annual wind rose and annually-averaged wind speed generated from the output of the Weather Research and Forecast model (www.wrf-model.org).

Gridded emissions of $SO_2$, $NO_x$, and $NH_3$ are obtained from the UK NAEI (http://naei.beis.gov.uk/) at 1 km × 1 km spatial resolution. Input emissions of $SO_2$ and $NO_x$ are split into three categories: UK area, point source, and shipping emissions. FRAME treats $SO_2$ emissions as 95% $SO_2$ and 5% $H_2SO_4$, and $NOx$ emissions as 95% $NO$ and 5% $NO_2$. For $NH_3$ emissions there are only UK area and point source categories. The $NH_3$ emissions from livestock are distributed spatially according to Hellsten et al. (2008). All emissions are injected into the air column at different heights according to the classification of

emission sources.

The chemical scheme includes gaseous and aqueous-phase oxidation reactions and conversion of the gases $NH_3$, $SO_2$, and $NO_x$ to particulate matter ($NH_4^+$, $NO_3^-$, $SO_4^{2-}$). Modelled dry deposition is land-cover dependent and calculated using a canopy resistance model. Wet deposition is calculated using scavenging coefficients and it is driven by rainfall, which is modelled using a constant drizzle approach based on the measured spatial distribution of annual average rainfall data with the assumption

of an enhanced washout rate over elevated areas.

A detailed evaluation of model outputs with annually averaged measurements of pollutant concentrations in air and precipitation concentrations is discussed elsewhere (Dore et al., 2015). In this study, all model runs were performed using emissions and meteorology data for the year 2012 and FRAME model version 9.15.0.



## 2.2 Sensitivity and uncertainty analysis

For both sensitivity and uncertainty analyses a Latin hypercube sampling design was chosen as it is superior to quasi-random sampling for small numbers of samples (Saltelli et al., 2008). A uniform LHS design was created using the R package 'lhs'(Carnell, 2016), with the sample optimised by maximising the mean distance between the design points. The LHS design

was created for the scaling coefficients applied to the model input emissions of UK $SO_2$, $NO_x$ and $NH_3$ and not for the actual values of the input emissions. This means that emissions from all sources of a particular pollutant were varied by the same fraction across all grid squares in a particular model run.

For the sensitivity analysis a uniform LHS sample of size $n = 100$ within a range of ± 40 % relative to the baseline for each of the three input variables was created. This range was chosen as it encompasses the range of variations in input emissions used

for future scenario simulations with the FRAME model.

Regression coefficients (RC) were used as the measure of the sensitivity of the model response, derived as follows. For each model grid cell, and for each model output variable, a multiple linear regression (Eq. 1) was performed using the data from the $n = 100$ model runs. To obtain the RCs ($b_i$ in Eq. 1) the model inputs $X_i$, and outputs $Y$, were substituted by corresponding values of fractional change relative to the baseline value. This simplifies interpretation of the resulting RCs. A RC represents

the relative effect of changing input $X_i$ on the output $Y$, e.g. RC = 0.5 signifies a 15 % reduction in the output variable value if an input is reduced by 30 %. The coefficients of determination ($R^2$) were evaluated for each fitted model to identify if a significant level of non-linearity in the input-output relationship was present.

$$Y = b_0 + \sum_{i=1}^{3} b_i X_i \tag{1}$$

For the uncertainty propagation, the input sampling space was constrained to the specific uncertainty ranges assigned to the emissions of $SO_2$, $NO_x$ and $NH_3$ in the UK Informative Inventory Report (Misra et al., 2015) with again a sample $n = 100$.

These uncertainty ranges are derived following published guidelines on quantifying uncertainties in emissions estimates (IPCC, 2006; Pulles and Kuenen, 2016). According to the guidelines, uncertainties are expressed as lower and upper limits of the 95 % confidence interval as a percentage of the central estimate. The assigned emissions uncertainties have ± 4 %, ± 10 % and ± 20 % ranges, for $SO_2$, $NO_x$ and $NH_3$ respectively. The probability distributions were not specified, therefore it was chosen to use uniform distributions for the variable ranges from which the LHS sample was created.

The uncertainty values for each grid square were calculated as a half of the 95% confidence interval relative to the mean value of the output as recommended in the EMEP/EEA and IPCC Guidebooks (IPCC, 2006; Pulles and Kuenen, 2016). Relative uncertainty values are presented here.

To assess the contribution of uncertainties in the emissions of $SO_2$, $NO_x$ and $NH_3$ to the overall output uncertainty standardised regression coefficients (SRCs) were calculated as shown in Eq. 2. A multiple linear regression was performed using the data

from the 100 model simulations for the case of constrained input sampling space. The SRCs ($\beta_i$ in Eq. 2) were calculated by





multiplying the RC by the ratio between the standard deviations of the input $\sigma_i$, and output $\sigma_Y$. ($\sigma_Y$ is the same for all the $\beta_i$ values for a given output variable.)

$$\beta_i = b_i \, \frac{\sigma_i}{\sigma_Y} \tag{2}$$

The squared value of SRC (Eq. 3) for linear additive models is equal to the ratio of variance of mean of $Y$ when one input variable is fixed, $V_{X_i}(E_{X_{\sim i}}(Y|X_i))$, to the unconditional variance of $Y$, $V(Y)$ (Saltelli et al., 2008). Thus SRC squared represents

the fractional contribution of the uncertainties in the model inputs to the overall uncertainty in the output. For the case of non-linear models, variance decomposition methods are described in more detail elsewhere (Homma and Saltelli, 1996; Saltelli, 2002; Sobol', 1993).

$$\beta_i^2 = \frac{V_{X_i}(E_{X_{\sim i}}(Y|X_i))}{V(Y)} \tag{3}$$

## 3 Results and discussion

### 3.1 Global sensitivity analysis

Figure 1 summarises the distributions of the regression coefficient (RC) global sensitivity measure across all model grid cells. RCs show the sensitivity of each model output variable to the three input emissions variables ($SO_2$, $NO_x$ and $NH_3$), and can be interpreted as a magnitude of the response of an output to the unit change in a particular input when all other inputs are allowed to vary. The magnitude of the RCs provides useful information not only about the effect of the change in a particular input on a model output, but also allows input sensitivity ranking to be determined because all inputs were assigned the same range of

variation (± 40 %). In the case where the ranges for inputs differ, standardised regression coefficients (SRCs) are used to obtain the input importance ranking instead.

Figure 1 shows that model outputs have (i) varying sensitivities, (ii) varying relative rankings in their sensitivities to $SO_2$, $NO_x$ and $NH_3$ emissions, and (iii) that these output sensitivities to the emissions also vary spatially across the model grids, as shown by the spreads in individual box plots. The annual average concentrations of particulate $NH_4^+$, $NO_3^-$, and $SO_4^{2-}$ and annual dry

and wet deposition of $SO_y$ for the baseline model run are presented in Supplementary Information Figure S1. The actual spatial distributions of the RCs from Figure 1 are illustrated in Figure 2 for the example output variables of particulate $NH_4^+$, $NO_3^-$, and $SO_4^{2-}$. Figure 3 shows the equivalent for the example output variables of dry and wet deposition of $SO_y$. These five output variables were chosen to illustrate the spatial distribution of uncertainty and sensitivity metrics. Figures S2 and S3 in Supplementary Information show the spatial distribution of RCs for other FRAME outputs displayed in Figure 1.

RC is a first-order sensitivity measure, also known as a main effect, and it quantifies the effect of varying a model input $X_i$ alone. In this study no second, or higher, order interaction terms were quantified as their contribution was assumed to be negligible. This was concluded from the values of the coefficients of determination ($R^2$) obtained from multiple linear regressions performed; for most output variables, values of $R^2$ were on average > 0.98 with the exception of a slightly lower



value for $HNO_3$ ($R^2 > 0.96$). Hence less than 2 % (4 % for $HNO_3$) of variance in the output could not be explained by the linear combination of inputs. This finding allows us to conclude that the FRAME model response is in fact fairly linear within the ± 40 % emission perturbation range investigated. The absence of any substantial deviations from linearity in the model response and absence of second or higher order interactions between input variables indicate that the current use of the FRAME

5    model to produce source-receptor matrices for the use in the UK Integrated Assessment Model is not subject to undue error from varying emissions one-at-a-time. Without conducting the global sensitivity analysis it is not possible to predict a priori for a given model output variable either the relative sensitivities to the different input factors, such as emissions, or the spatial variation in these sensitivities that are illustrated in Figures 1, 2 and 3.

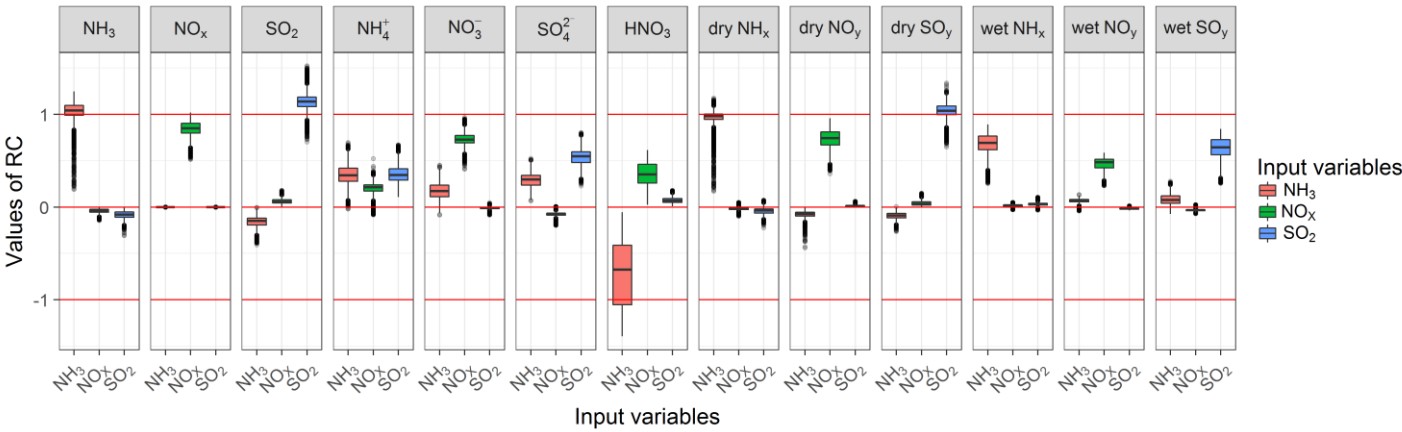

10    **Figure 1 Box plots of the values of regression coefficients (RC) across all UK land-based model grid squares. Boxes demarcate the median and lower/upper quartiles of the distributions; whiskers extend to 1.5 times the interquartile range.**





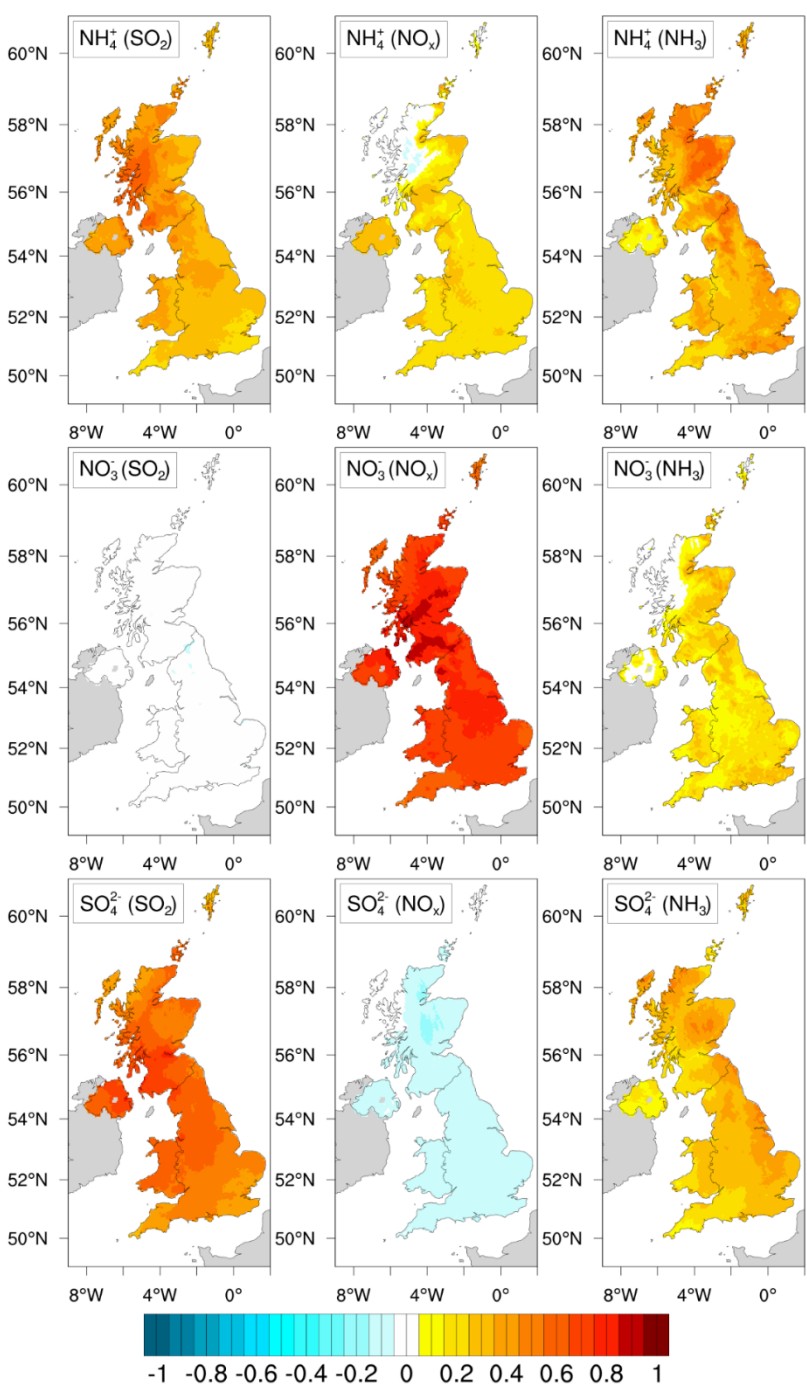

**Figure 2 Spatial distributions (at the 5 km × 5 km model grid resolution) of RCs for particulate NH₄⁺, SO₄²⁻, and NO₃⁻ as a function of variation in input emissions of SO₂, NOₓ or NH₃. The model input emissions for which the RC quantifies the output variable sensitivity is given in brackets in each panel.**





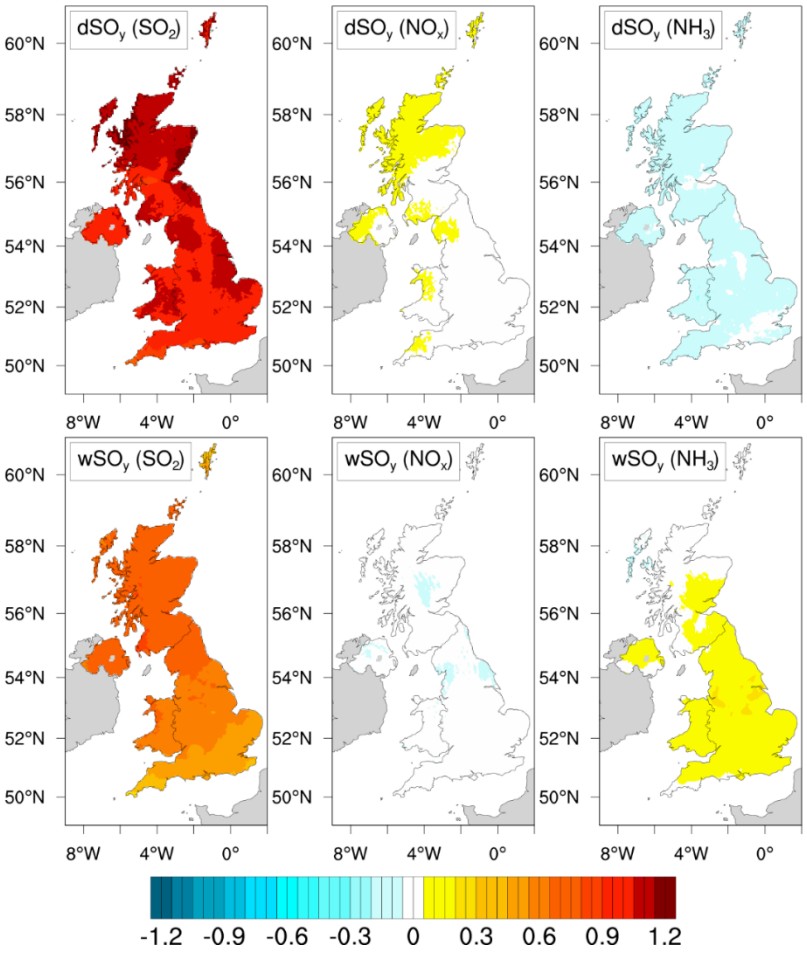

**Figure 3 Spatial distributions (at the 5 km × 5 km model grid resolution) of RCs of dry (d) and wet (w) deposition of $SO_y$ as a function of variation in input emissions of $SO_2$, $NO_x$ or $NH_3$. The model input emissions for which the RC quantifies the output variable sensitivity is given in the brackets in each panel.**

With respect to findings from this FRAME model sensitivity analysis for particulate inorganic components in the UK context, Figure 1 shows that the modelled surface concentrations of particulate $NH_4^+$ are sensitive to changes in emissions of all three pollutants, being similarly sensitive (on average) to emissions of $NH_3$ and $SO_2$, and slightly less sensitive to emissions of $NO_x$. The sensitivities of $NH_4^+$ to $SO_2$, $NO_x$ and $NH_3$ emission changes were found to vary substantially around the UK (top row of

10    Figure 2). Sensitivity of $NH_4^+$ to $SO_2$ emissions is generally lowest in south-east England, and rises on moving north and west across the UK. Reductions in emissions are always associated with reductions in $NH_4^+$. The broad geographical pattern of relative sensitivity across the UK of $NH_4^+$ to $NH_3$ emissions is approximately the reverse of that to $SO_2$ emissions although with substantial spatial heterogeneity as well. Figure 2 shows that there are instances in north-west Scotland of negative RCs for the sensitivity of $NH_4^+$ to $NO_x$ emissions, i.e. areas where $NH_4^+$ increases when $NO_x$ emissions are decreased.



Figure 1 similarly shows that surface concentrations of particulate $SO_4^{2-}$ are sensitive to changes in emissions of all three of $SO_2$, $NO_x$ and $NH_3$ (most sensitive to $SO_2$ emissions) but with a universally negative sensitivity (albeit relatively weak) to $NO_x$ emissions, i.e. particulate $SO_4^{2-}$ concentrations increase everywhere by approximately 3 % if $NO_x$ emissions are reduced by 40 % (lower row of Figure 2). This is due to competition between $HNO_3$ and $H_2SO_4$ to react with $NH_3$ and form particles, i.e.

reducing $NO_x$ emissions means $NH_3$ is more readily available to react with $H_2SO_4$. The positive values of RCs of $SO_4^{2-}$ to $SO_2$ emissions are geographically fairly uniform (somewhat lower sensitivity in the eastern UK), but the relative sensitivity to $NH_3$ emissions is more heterogeneous and greater in the east.

The sensitivity of particulate $NO_3^-$ concentrations to the emissions is more straightforward than for particulate $NH_4^+$ and $SO_4^2$, being dominated by its positive sensitivity to $NO_x$ emissions, weakly sensitive to $NH_3$ emissions and essentially not sensitive

at all to $SO_2$ emissions (Figure 1 and middle row of Figure 2). The sensitivity to $NO_x$ emissions is almost unity, such that for example a 30 % reduction in $NO_x$ emissions results in almost the same 30 % reduction in surface $NO_3^-$. The spatial distribution of RCs that represent sensitivity of $NO_3^-$ concentrations to $NO_x$ (and $NH_3$) emissions is also geographically more homogenous across the UK than the sensitivities of $NH_4^+$ and $SO_4^{2-}$ concentrations (middle row of Figure 2).

The concentrations of the three inorganic particulate matter components are determined by the reactions that lead to formation

of $(NH_4)_2SO_4$ and $NH_4NO_3$. Formation of the former is irreversible whilst the latter exists in reversible equilibrium with gas-phase $NH_3$ and $HNO_3$. Changes in emissions of $NH_3$ have an impact on formation of both $(NH_4)_2SO_4$ and $NH_4NO_3$ very quickly, and therefore close to the source of the $NH_3$ emissions, because it reacts directly as $NH_3$. In contrast the influence of changes in $SO_2$ and $NO_x$ emissions is not so localised. Before they influence the formation of $(NH_4)_2SO_4$ and $NH_4NO_3$ these gases must be oxidised in the atmosphere to $H_2SO_4$ and $HNO_3$, during which time the air is undergoing transport. The spatial

pattern of the sensitivities of $(NH_4)_2SO_4$ and $NH_4NO_3$ formation to changes in the UK precursor emissions is therefore the outcome of many interacting factors: i) the magnitude of background import of precursors from outside the UK which could explain lower sensitivity of inorganic particulate matter components to $SO_2$ emissions in south-east England, ii) the magnitude and spatial pattern of the UK precursors, iii) the time taken for chemical oxidation in relation to atmospheric transport of air masses, and iv) the varying dry and wet deposition spatial patterns that remove from the atmosphere both the precursor gases

and particulate products.

In summary, the broad patterns of the sensitivity results in Figures 1, 2 and 3 can be explained as follows. The surface concentrations of the directly emitted pollutants $NH_3$, $NO_x$ and $SO_2$ are predominantly sensitive only to their respective emissions (Figure 1). This is also the case for the deposition of oxidised S, and of oxidised and reduced N. Dry deposition is dominated by the gas-phase components so the variations in the dry deposition of $NH_x$ and $SO_y$ are dominated by the variations

in the emissions of $NH_3$ and $SO_x$ respectively with the RC values being close to 1. For the dry deposition of $NO_y$, both $NO_2$ and its oxidation product $HNO_3$ are important. This is illustrated by the weaker response of dry $NO_y$ deposition to changes in $NO_x$ emissions. Wet deposition is a more complex process as this is dominated by washout of the particles which are the product of chemical reactions in the atmosphere. This explains lower values of RC for wet compared to dry deposition.





The considerably more ubiquitous sources of $NO_x$ emissions compared with $SO_2$ emissions means that atmospheric concentrations of gaseous oxidised N are generally higher than for oxidised S so the former usually has greater influence on $NH_3$ chemistry. Therefore particulate $NO_3^-$ is predominantly controlled by $NO_x$ emissions, and changes in $SO_2$ emissions have very little effect on particulate $NO_3^-$. However, because lower $NO_x$ emissions lead to lower $NH_4NO_3$ formation more $NH_3$ is

available which means lower $NO_x$ emissions leads to greater $(NH_4)_2SO_4$ formation this explains the inverse correlation between surface concentrations of $SO_4^{2-}$ and $NO_x$ emissions. On the other hand, changes in $NH_3$ emissions impact on both $NO_3^-$ and $SO_4^{2-}$ concentrations, both in a positive direction of association, but with a magnitude sensitive to the relative amounts of the reacting species present, which in turn depends both on the magnitudes and distances of local sources and on long-range transport. Likewise, the sensitivity of $NH_4^+$ concentrations varies with all three sets of precursor emissions and with

geographical location. The same is the case for concentrations of $HNO_3$. This is why, aside from some broad expectations, it is not easily possible to predict the spatial patterns of the sensitivities of ACTM model output to changes in emissions and a formal sensitivity analysis is needed.

### 3.2 Uncertainty propagation

The global uncertainty propagation approach for FRAME output variables was based on the assigned uncertainties in the

estimates of the total UK emissions of $SO_2$ (± 4 %), $NO_x$ (± 10 %) and $NH_3$ (± 20 %) (Misra et al., 2015). As explained in the Methods, the uncertainties in the input emissions were assigned uniform distributions. No substantial difference in the resulting model output uncertainty ranges was observed when the probability distributions of the input emissions were changed to normal. The distributions of the relative uncertainties across all model grid cells for each output are shown in Figure 4. Example maps of the spatial distributions of the relative uncertainties from Figure 4 for surface concentrations of particulate $NH_4^+$, $NO_3^-$

, and $SO_4^{2-}$ and for dry and wet deposition of $SO_y$ are shown in Figure 5. Equivalent maps for the relative uncertainties of the other FRAME output variables are shown in Supplementary Information Figure S4.



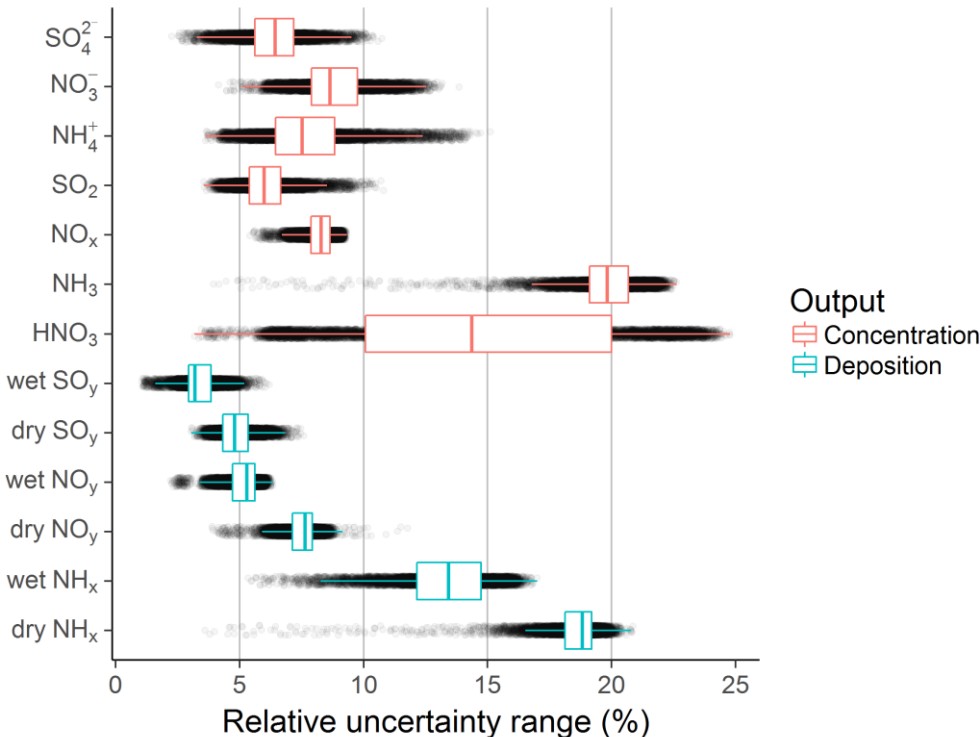

**Figure 4 Distributions of relative uncertainty values calculated for all FRAME model outputs across all model grid squares given the following input uncertainty ranges: ± 4 %, ± 10 % and ± 20 % in emissions of $SO_2$, $NO_x$ and $NH_3$ respectively. Boxes demarcate the median and lower and upper quartiles of the distributions; whiskers extend to 1.5 times the interquartile range.**

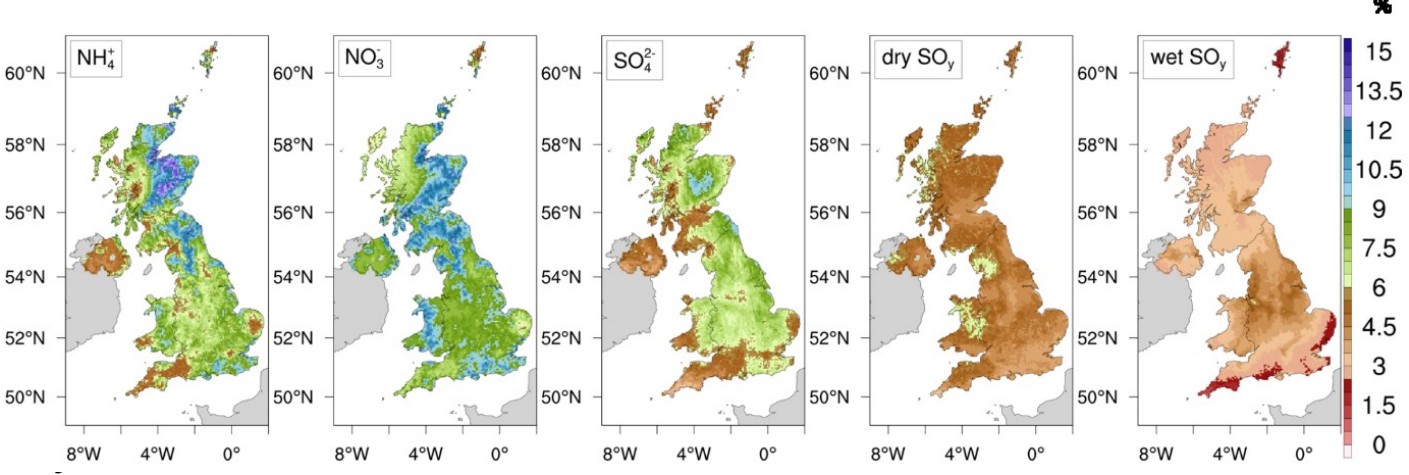

**Figure 5 Spatial distributions (at the 5 km × 5 km model grid resolution) of the relative uncertainties in surface concentrations of particulate $NH_4^+$, $SO_4^{2-}$, and $NO_3^-$ and dry and wet deposition of $SO_y$ for uncertainties of ± 4 %, ± 10 %, ± 20 % in emissions of $SO_2$, $NO_x$ and $NH_3$ respectively. The uncertainty values are represented as a range of +/- the baseline value and represent the 95 % confidence interval.**



Figure 4 shows that the surface concentration of $NH_3$ is the most uncertain output (model grid median uncertainty 19.8 %). This is because the variation in $NH_3$ surface concentrations is almost entirely driven by variation in $NH_3$ input emissions (Figure 1) and this is the most uncertain input in the presented analysis. The uncertainty in modelled dry deposition of $NH_x$ likewise closely matches the assigned uncertainty in $NH_3$ emissions (median = 18.8 %). The uncertainty in wet deposition of

$NH_x$ is somewhat less than uncertainty in dry deposition (median = 13.4 %) because wet deposition of $NH_x$ includes some dissolved $(NH_4)_2SO_4$ component which is also sensitive to other precursor emissions whose uncertainty is estimated to be smaller than for $NH_3$. Surface concentrations of $SO_2$ and the dry and wet depositions of $SO_y$ have least uncertainty (medians of 6.0 %, 4.8 % and 3.2 %) for the similar reason that these model outputs are predominantly sensitive to $SO_2$ emissions (Figure 1) which has the smallest of the input uncertainties (± 4 %).

Relative uncertainties of particulate $SO_4^{2-}$ (median = 6.4 %), $NO_3^-$ (median = 8.6 %) and $NH_4^+$ (median = 7.5 %) are fairly similar (Figure 4) even though there are substantial differences in the assigned uncertainties for emissions of $SO_2$, $NO_x$ and $NH_3$. The explanation is that PM components are sensitive to all three inputs (for $NO_3^-$ two out of three inputs) (Figure 1). There is also wide spatial variation in the uncertainties of these PM components (Figures 4 and 5). The relative uncertainty values in surface concentration of $HNO_3$ show the largest variability out of all output variables. This can be explained by the

fact that the concentration of this species is impacted directly by both gas and particle-phase processes. The spatial pattern of the relative uncertainty values does not correlate either with the spatial pattern of emissions or rainfall, which demonstrates again that the uncertainties of many model outputs cannot be readily predicted because of the complexity of the atmospheric processes underpinning them and consequently that formal uncertainty analysis needs to be applied.

### 3.2.1 Uncertainty apportionment

Estimated uncertainty of the model output given the uncertainties in model input emissions is presented in Figures 4 and 5, but it is also of interest to know how each of the inputs contributes to the overall uncertainty individually. This was estimated by calculating squared standardised regression coefficients (SRCs) (Eq. 3). As an example, Figure 6 illustrates the spatial distributions of the fractional contributions of the $SO_2$, $NO_x$ and $NH_3$ emission uncertainties to the overall uncertainties in surface concentrations of particulate $NH_4^+$, $NO_3^-$ and $SO_4^{2-}$, for the assigned uncertainties in the input emissions, whilst Figure

7 illustrates similar for the dry and wet deposition of $SO_y$. The equivalent maps for the other model output variables are presented in Supplementary Information Figures S5 and S6.

Figure 6 shows that across nearly all the UK, uncertainty in concentrations of particulate $NH_4^+$ is mainly driven by the uncertainty in $NH_3$ emissions. Uncertainty in $NO_x$ emissions contributes some uncertainty to $NH_4^+$ concentrations, whilst the uncertainty in $SO_2$ emissions makes almost no contribution. Northern Ireland is an exception; here uncertainties in $NO_x$

emissions contribute the most to the uncertainties in $NH_4^+$ concentrations and perturbations in $NH_3$ emissions have less impact. Concentrations of $NH_3$ in Northern Ireland are some of the highest anywhere in the UK, whilst $NO_x$ emissions are not high; this means that $NH_3$ will be in excess so the formation of $NH_4NO_3$ will be largely controlled by $HNO_3$ through $NO_x$ emissions. The major contribution to uncertainty in particulate $NO_3^-$ derives from uncertainty in $NO_x$ emissions (Figure 6). However in





the east of Scotland, uncertainty in $NH_3$ emissions contributes up to 78% of the total uncertainty. There is no contribution from $SO_2$ emissions uncertainty. An important feature of the lower panel of Figure 6 is that by far the major contributor to uncertainty in particulate $SO_4^{2-}$ concentrations is the uncertainty assigned to the $NH_3$ emissions not the uncertainty in the direct precursor $SO_2$ emissions. This is because the formation of $(NH_4)_2SO_4$ is irreversibly dependent on gaseous $NH_3$ and emissions of $NH_3$

5     are much more uncertain than $SO_2$ emissions.

Figure 7 shows the spatial distribution of the squared SRC values for dry and wet $SO_y$ deposition; for these output variables uncertainty in $NO_x$ does not make any contribution to uncertainty in either case. In contrast to the situation for particulate $SO_4^{2-}$ concentrations shown in Figure 6, Figure 7 shows that uncertainty in dry and wet deposition of $SO_y$ is mainly driven by the uncertainty in the $SO_2$ emissions. Additionally uncertainty in $NH_3$ emissions contributes to the total uncertainty in dry and wet

10    $SO_y$ deposition. The contribution to uncertainty in wet deposition is higher due to wet deposition being dominated by the washout of the particles which include products of the reactions of $NH_3$ with oxidation products of $SO_x$.





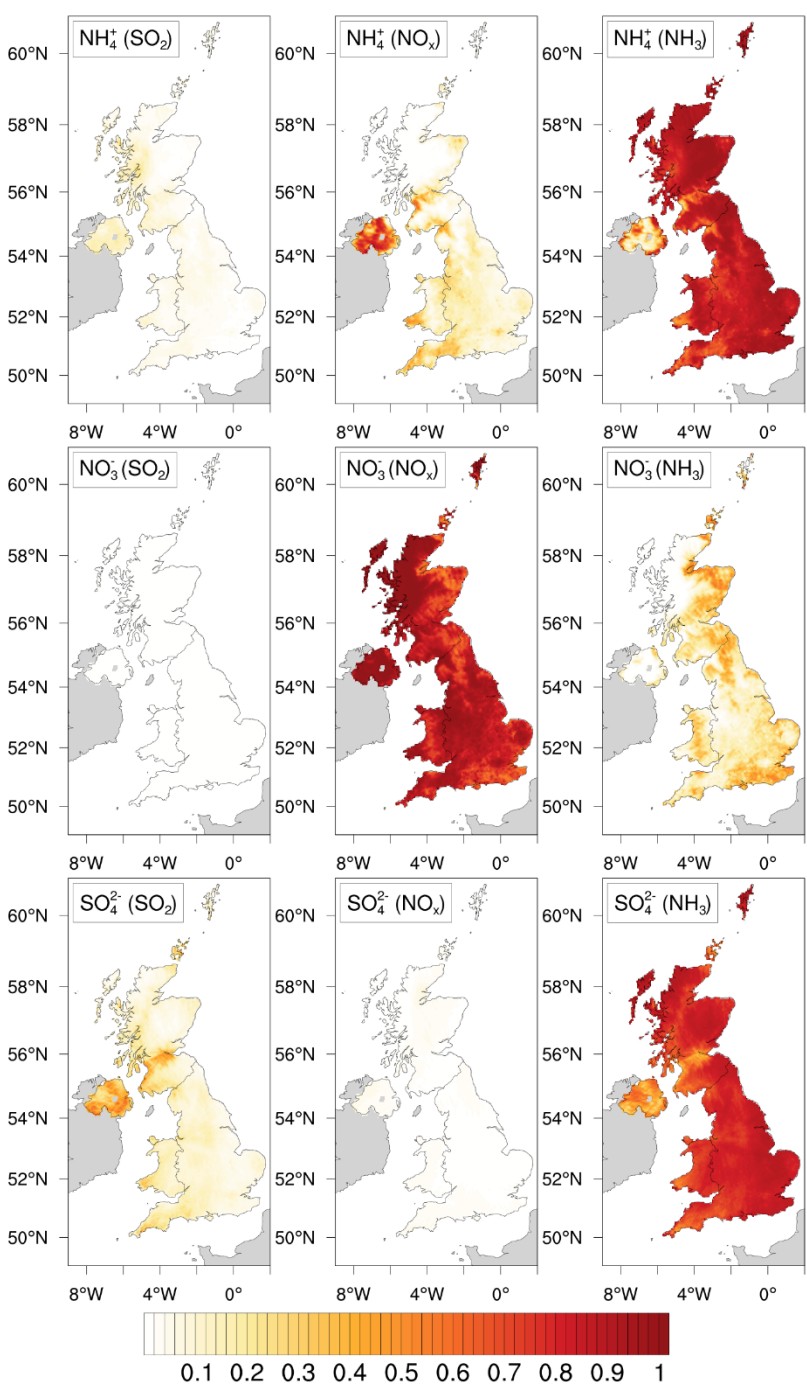

**Figure 6 Spatial distributions (at the 5 km × 5 km model grid resolution) of the squared SRC values which represent the fractional contribution of the uncertainty in the input emissions given in brackets to the overall uncertainty in the surface concentrations of particulate $NH_4^+$, $SO_4^{2-}$, and $NO_3^-$. The uncertainties in the input emissions are ± 4 %, ± 10 % and ± 20 % for $SO_2$, $NO_x$ and $NH_3$ respectively.**



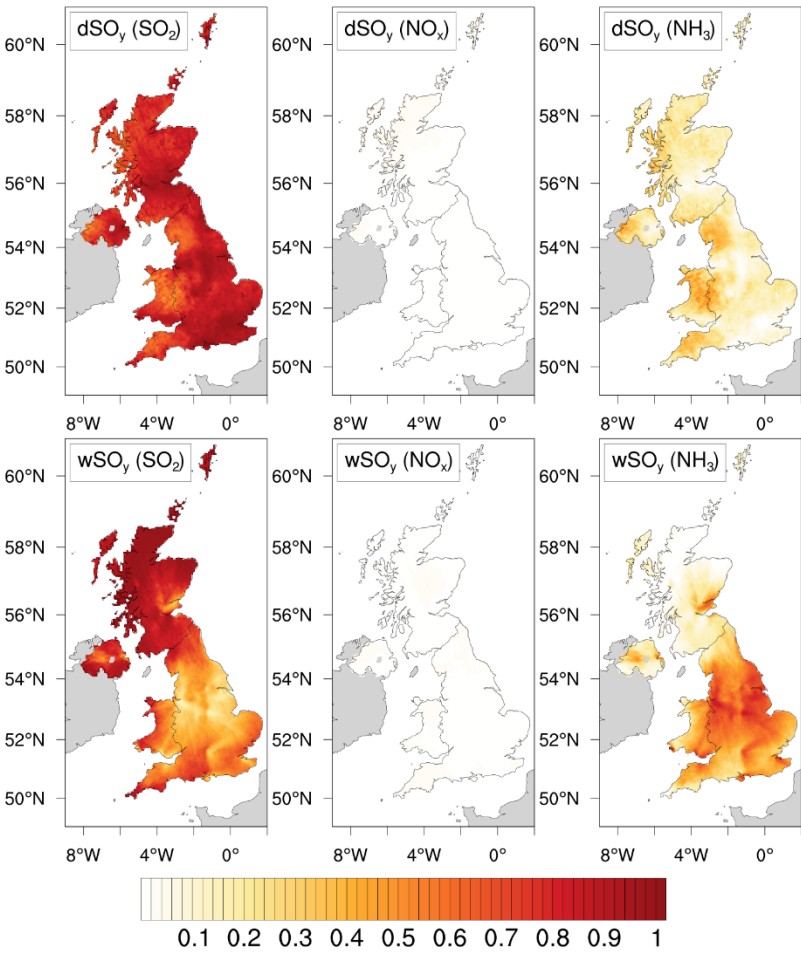

**Figure 7 Spatial distributions (at the 5 km × 5 km model grid resolution) of the squared SRC values which represent the fractional contribution of the uncertainty in the input emissions given in brackets to the overall uncertainty in the dry and wet deposition of SO_y. The uncertainties in the input emissions are ± 4 %, ± 10 % and ± 20 % for SO₂, NOₓ and NH₃ respectively.**

**4 Conclusions**

We have applied global sensitivity analysis to determine the response of concentration and deposition output variables of the FRAME atmospheric chemistry transport model to perturbations of UK emissions of $SO_2$, $NO_x$ and $NH_3$. The benefit of using systematic global sensitivity analysis is that all dimensions of variable input space are investigated simultaneously, which is important when the response to a large number of variables is of interest, so inferences can be drawn without assumptions about the model structure. For complex models such as ACTMs, for which input-output mapping is not analytically tractable,




it is not possible to predict output sensitivities to multiple input perturbations without conducting a global sensitivity analysis. Local one-at-a-time sensitivity analysis is often applied without acknowledging the shortcomings associated with it.

In this study no substantial deviations from linearity or presence of interactions between the model input variables were identified for the FRAME model in response to input emission perturbations within a ±40 % range, hence regression

coefficients obtained from multiple linear regression were chosen as a sensitivity measure. This was not predictable from a local one-at-a-time sensitivity analysis.

Whilst sensitivity of surface concentrations of the primary precursor gases $SO_2$, $NO_x$ and $NH_3$ (and of deposition of S and N) was dominated by the emissions of the respective pollutant, the sensitivities of secondary species such as $HNO_3$ and particulate $SO_4^{2-}$, $NO_3^-$ and $NH_4^+$ to pollutant emissions were more nuanced and geographically variable. The dry deposition of S and N

showed stronger response to changes in the emissions of the respective pollutant compared to wet deposition.

A global uncertainty analysis approach was used to estimate uncertainty ranges for all FRAME model output variables from the uncertainties assigned to the UK emissions of $SO_2$, $NO_x$ and $NH_3$ (± 4 %, ± 10 % and ± 20 % respectively) by the UK National Atmospheric Emissions Inventory. The spatial distribution of the relative uncertainty was affected by both the sensitivity of the model output to variations in the inputs and the magnitude of this variation (i.e. the input uncertainty range);

$NH_3$ was the most uncertain input and as a result the output variables sensitive to $NH_3$ showed the highest levels of relative uncertainty in the areas most sensitive to this input. The uncertainty in the surface concentrations of $NH_3$ and $NO_x$ and the depositions of $NH_x$ and $NO_y$ was shown to be due to uncertainty in a single precursor input variable, $NH_3$ and $NO_x$ respectively. In contrast, the concentration of $SO_2$ and deposition of $SO_y$ was affected by uncertainties in both $SO_2$ and $NH_3$ emissions. Likewise, the relative uncertainties in the modelled surface concentrations of each of the secondary pollutant variables ($NH_4^+$,

$NO_3^-$, $SO_4^{2-}$, and $HNO_3$) were affected by the uncertainty range of at least two input variables.

This work has demonstrated a methodology for conducting global sensitivity and uncertainty analysis for ACTMs. Although, for the FRAME model used here, the response to emission perturbations was found to be substantially linear in the investigated input range, the complexity of chemical and physical processes included in ACTMs means that the input-output relationships, in particular their spatial patterns, cannot be predicted without conducting a global sensitivity analysis. The benefit of using

global approaches is that all dimensions of input variable space are investigated simultaneously so model input-output relationships can be quantified without the need to make strong prior assumptions about the model response to perturbations in the inputs of interest.

**Data availability**

The FRAME model code is not available in the public domain as the model is the intellectual property of the Centre for

Ecology & Hydrology and is only made available to students and researchers who are collaborating directly with CEH staff. However, all the following output data are available at: https://doi.org/10.5281/zenodo.1145852.

1) All FRAME model outputs (raw data) for both actual input uncertainty and ±40 % input ranges.

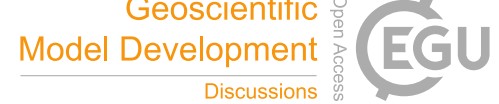

2) R scripts used to calculate RCs, SRCs, and uncertainty ranges.

3) Calculated RCs, SRCs, and uncertainty ranges for every FRAME output variable, which underpin all figures in this paper.

**Competing interests**

The authors declare they have no competing interests.

**Acknowledgements**

K. Aleksankina acknowledges studentship funding from the University of Edinburgh and the NERC Centre for Ecology &
Hydrology (NERC CEH project number NEC05006). The CEH funding was provided by the Department for Environment
Food & Rural Affairs contract AQ0947 Support for National Air Pollution Control Strategies 2013-2015 (SNAPS).

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
