# Peer review of "Global sensitivity and uncertainty analysis of an atmospheric chemistry transport model: the FRAME model (version 9.15.0) as a case study"

_Geoscientific Model Development, 2017_

## Referee Comment (RC1) · Anonymous Referee #1 · 15 Feb 2018

This paper presents a very useful approach for quantification of the impact of emissions uncertainty on modelled concentrations and deposition of sulphur and nitrogen species. The material is presented clearly and the conclusions are supported by the results presented.

I have a few minor comments about the methods section, where I think some further details would be useful.

The annual average wind rose and wind speed used to calculate trajectories in the FRAME model are generated from WRF - what period was used to generate these trajeories, what resolution was WRF run at, what version of WRF was used and what

meteorology was used to drive WRF at the boundaries?

More detail of the inorganic chemistry scheme in FRAME and information on the type of inorganic aerosol module used, with references for both of these.

The approach taken to the representation of the emission uncertainty (varying the emissions in all grid boxes by the same fraction in each run) is justified in the context of this study. However, it does mean that several important aspects of emissions uncertainty are not included. In particular any uncertainties in the spatial distribution or height of emission are not captured. There are also important sources of emission related uncertainty that FRAME cannot capture such as uncertainty in diurnal or seasonal cycles of input. These limitations should be noted here.

Finally, it would be interesting to see the impact of these uncertainties on the secondary inorganic aerosol mass. This may be beyond the scope of this study, but concentrations of PM2.5 are highly relevant for air quality forecasting and policy relevant research. If the results are available, it would be a valuable addition to this study.

---

## Referee Comment (RC2) · Anonymous Referee #2 · 21 Feb 2018

The paper seeks to estimate the uncertainty and sensitivity of multiple atmospheric chemistry transport model output with respect to three uncertain inputs. The authors use optimised Latin hypercube design to sample from the computationally expensive computer model and use three different methods to summarise the uncertainty and sensitivity of various outputs to the uncertain inputs.

In general I found the paper well written and easy to follow but I don't follow the reasoning of the two sensitivity measures and the difference between the two. My concerns lie in the choice of methods used to assess the uncertainty and sensitivity in the outputs and my comments are focussed in this direction.

[Figure]

Method 1 - RC

The first method uses regression to estimate the coefficients of a linear model in order to assess output sensitivity to each model input.

Regression is not considered to be a particularly good way to estimate global sensitivity measures (since they are not very robust) and the 'main effects' that the authors refer to would normally be associated with a variance-based sensitivity analysis. Can the authors say more about why they feel this is a more appropriate method to use than variance-based sensitivity or what they are trying to capture that is different? In any case, I don't think the authors should use the term 'main effects' for regression coefficients due to their common use elsewhere.

A 100 point Latin hypercube design has been used to vary the parameters within +/- 40% for the regression. I don't understand the reasoning behind these ranges when this is way beyond those considered plausible by the UK Informative Inventory. Can the authors justify this better and say why the regression doesn't follow the emissions uncertainties?

It is recognised that the regression coefficients are only likely to be meaningful if the model is linear, as measured here by $R^2$. Has $R^2$ been calculated for all model gridboxes? It's not clear from the reporting of the value that it is calculated everywhere – I assume it must be as their needs to be a regression model at every grid box. How big does $R^2$ need to be for the regression coefficients to be useful?

What happens to the regression coefficients when the intercept term is not included in the model?

In the text line 12 it is stated that the RC 'can be interpreted as the response of an output to a unit change in a particular input when all others are allowed to vary' but in line 25 'RC quantifies the effect of varying a model input $X_i$ alone'. These are contradictory and line 25 is a better description of method 3 (although this section is

discussed later).

Method 2 – uncertainty propagation

The second method propagates the uncertainty in the emissions to the output using the estimated uncertainties from the UK Informative Inventory Report. Please make it clearer that a new sample has been created here.

On line 25 the authors state that the uncertainty is calculated as half the 95% CI relative to the mean value. Half the 95% CI gives $2\sigma$ – why is this used as opposed to $\sigma$? Can't $\sigma$ be calculated directly from the data as I assume it is used to calculate the 95% CI in the first place? It would be helpful to see the emissions maps – even if just in the supplementary material.

Method 3 - SRC

I need some more convincing that the SRCs calculated here are the same as the measures from Saltelli 2008 – can you expand this? It is these measures that are normally be referred to as the main effects.

What is the $R^2$ for these new regression fits and how were $sigma\_i$ and $sigma\_Y$ derived? $sigma\_i$ is stated as the standard deviation of the input – it should be the standard deviation of the output given uncertainty in the input.

I would also expect the measures here to follow the regression coefficients more closely given the linearity in the model. This measure is giving different information to the RC and I don't fully understand what that difference is and why the results are different.

I think the authors should consider using generalised additive modelling here to calculate the main effects following Strong M, Oakley JE, Brennan A. Estimating multi-parameter partial Expected Value of Perfect Information from a probabilistic sensitivity analysis sample: a non-parametric regression approach. Medical Decision Making. 2014;34(3):311-26 particularly Eq 6 and 8.

The references should be expanded to include other uses of sensitivity analysis in earth science models. These tools are generally applicable across different types of models which is an important point to make.
* * *

---

## Author Comment (AC1) · 12 Mar 2018

**gmd-2017-302: Global sensitivity and uncertainty analysis of an atmospheric chemistry transport model: the FRAME model (v. 9.15.0) as a case study**
**by Aleksankina et al.**

**Response to reviewer #1**

*This paper presents a very useful approach for quantification of the impact of emissions uncertainty on modelled concentrations and deposition of sulphur and nitrogen species. The material is presented clearly and the conclusions are supported by the results presented. I have a few minor comments about the methods section, where I think some further details would be useful.*

Response: We thank the reviewer for their very positive comments on the usefulness and presentation of our work.

*(1) The annual average wind rose and wind speed used to calculate trajectories in the FRAME model are generated from WRF - what period was used to generate these trajectories, what resolution was WRF run at, what version of WRF was used and what meteorology was used to drive WRF at the boundaries?*

Response: The following expanded text and additional citation has now been added to the end of the first paragraph of Section 2.1.
"The trajectories are defined by an annual wind rose and annually-averaged wind speed generated for year 2012 from the output of the Weather Research and Forecast model (www.wrf-model.org) (Skamarock et al., 2008) version 3.7.1. The model was run at a 5 km resolution over the UK with boundary and initial conditions initialised by the National Centers for Environmental Prediction Final Global Forecast System (NCEP-GFS-FNL) data (https://rda.ucar.edu/datasets/ds083.2/)."

Skamarock, W. C., Klemp, J. B., Dudhia, J., Gill, D. O., Barker, D. M., Duda, M. G., Huang, X. Y., Wang, W. and Powers. J. G. (2008) A description of the advanced research WRF version 3. NCAR technical note NCAR/TN-475+STR, 10.5065/D68S4MVH.

*(2) More detail of the inorganic chemistry scheme in FRAME and information on the type of inorganic aerosol module used, with references for both of these.*

Response: The following expanded text and additional citation is now included in Section 2.1.
"The chemical scheme is described in Fournier et al. (2004) and includes gaseous and aqueous-phase oxidation reactions and conversion of the gases $NH_3$, $SO_2$, and $NO_x$ to particulate matter ($NH_4+$, $NO_3^-$, $SO_4^{2-}$). $NH_4NO_3$ is formed by the equilibrium reaction between $HNO_3$ and $NH_3$ and nitrate aerosol also arises by the deposition of $HNO_3$ onto sea salt or large particles. $H_2SO_4$ reacts with $NH_3$ to form $(NH_4)_2SO_4$. The aqueous phase reactions include the oxidation of S(IV) by $O_3$ and the metal catalysed reaction with $O_2$."

Fournier, N., Dore, A. J., Vieno, M., Weston, K. J., Dragosits, U. and Sutton, M. A. (2004) Modelling the deposition of atmospheric oxidised nitrogen and sulphur to the United Kingdom using a multi-layer long-range transport model. Atmos. Env. 38, 683-694.

*(3) The approach taken to the representation of the emission uncertainty (varying the emissions in all grid boxes by the same fraction in each run) is justified in the context of this study. However, it does mean that several important aspects of emissions uncertainty are not included. In particular any uncertainties in the spatial distribution or height of emission are not captured. There are also important sources of emission related uncertainty that FRAME cannot capture such as uncertainty in diurnal or seasonal cycles of input. These limitations should be noted here.*

Response: Thank you for these additional limitations we should highlight. The following text has been added to Section 2.2 where we describe the uncertainties in total annual emissions. "It is also acknowledged that a number of other aspects of emissions uncertainty are not included. For example, the FRAME model cannot capture uncertainty in assigned seasonal and diurnal cycles in emissions. Uncertainties in the spatial distributions or in height of elevated emissions are also not included."

An additional reminder of other emissions uncertainties has also been added at the start of Section 3.2 when presenting the results of the uncertainty propagation.

*(4) Finally, it would be interesting to see the impact of these uncertainties on the secondary inorganic aerosol mass. This may be beyond the scope of this study, but concentrations of PM2.5 are highly relevant for air quality forecasting and policy relevant research. If the results are available, it would be a valuable addition to this study.*

Response: We agree this is an important policy-relevant question. We used the FRAME model in this work as a 'proof of concept' for this global sensitivity approach. We are currently applying our methods to the more sophisticated EMEP4UK atmospheric chemistry transport model (www.emep4uk.ceh.ac.uk), which incorporates simulation of all PM components, including a more advanced formulation of the formation of secondary inorganic and organic aerosols, and VOC-NOx-ozone chemistry, and will be reporting on the findings from this model in other papers being prepared.

---

## Author Comment (AC2) · 12 Mar 2018

**gmd-2017-302: Global sensitivity and uncertainty analysis of an atmospheric chemistry transport model: the FRAME model (v. 9.15.0) as a case study**
**by Aleksankina et al.**

**Response to reviewer #2**

*The paper seeks to estimate the uncertainty and sensitivity of multiple atmospheric chemistry transport model output with respect to three uncertain inputs. The author use optimised Latin hypercube design to sample from the computationally expensive computer model and use three different methods to summarise the uncertainty and sensitivity of various outputs to the uncertain inputs.*
*In general I found the paper well written and easy to follow but I don't follow the reasoning of the two sensitivity measures and the difference between the two. My concerns lie in the choice of methods used to assess the uncertainty and sensitivity in the outputs and my comments are focussed in this direction.*

Response: We thank the reviewer for their time spent reviewing our paper and their positive comments on its presentation. We respond to the latter part of the comment as they arise point-by-point below.

*Method 1 - RC*
*(1) The first method uses regression to estimate the coefficients of a linear model in order to assess output sensitivity to each model input. Regression is not considered to be a particularly good way to estimate global sensitivity measures (since they are not very robust) and the 'main effects' that the authors refer to would normally be associated with a variance-based sensitivity analysis. Can the authors say more about why they feel this is a more appropriate method to use than variance-based sensitivity or what they are trying to capture that is different? In any case, I don't think the authors should use the term 'main effects' for regression coefficients due to their common use elsewhere.*

Response: We follow the suggested practices for global sensitivity analysis of Saltelli and Annoni (2010) who state that multiple linear regression is a suitable approach particularly if there is no substantial deviation from the linearity present in the model, as is the case for our FRAME model analysis here. We use variance-based sensitivity measures in the second part of our analyses where we investigate uncertainty apportionment, whereas in the first part of our work we were seeking information in overall trends of model response to changes in input emissions.

We agree that the terminology 'main effects' can be ambiguous. Therefore, as we had only referred to the term once in the paper we have removed it from that sentence, which now reads (p 7, line 7): "RC is a first-order sensitivity measure and it quantifies the average response of model output to varying a model input $X_i$ when all inputs are allowed to vary."

Saltelli, A. and Annoni, P.: How to avoid a perfunctory sensitivity analysis, Environ. Model. Softw., 25(12), 1508–1517, doi:10.1016/j.envsoft.2010.04.012, 2010.

*(2) A 100 point Latin hypercube design has been used to vary the parameters within +/-*

*40% for the regression. I don't understand the reasoning behind these ranges when this is way beyond those considered plausible by the UK Informative Inventory. Can the authors justify this better and say why the regression doesn't follow the emissions uncertainties?*

Response: The purpose for extending the range of variation for the emission input variables (beyond the range suggested by the reported uncertainties) was to test the overall model response to changes in emissions. Here the aim was to learn about the model; whether there is possibility of non-linearities or interaction terms being present in the model response. The range of +/- 40% was chosen because it encompasses the range of variations in input emissions used for future scenario simulations with the FRAME model, as well as incorporating emission reductions applied for the generation of source-receptor relationships for integrated assessment modelling. For example, linearity in the model response to emissions changes is assumed when estimating response to different scenarios, hence it is important to check that this assumption is valid when emissions are varied within a certain range from their nominal value. The sentence in the Methods section has been extended as follows (p5 line 12): "This range was chosen to test the overall model response to changes in emissions (for example to identify non-linearities) as it encompasses the range of variations in input emissions used for future scenario simulations with the FRAME model, as well as incorporating emission reductions applied for the generation of source-receptor relationships for integrated assessment modelling."

*(3)  It is recognised that the regression coefficients are only likely to be meaningful if the model is linear, as measured here by R^2. Has R^2 been calculated for all model gridboxes? It's not clear from the reporting of the value that it is calculated everywhere – I assume it must be as their needs to be a regression model at every grid box. How big does R^2 need to be for the regression coefficients to be useful?*

Response: Yes, $R^2$ was calculated for all grid cells, as specified in the following sentences in the Methods section (p5, line 16): "For each model grid cell, and for each model output variable, a multiple linear regression was performed using the data from the $n = 100$ model runs."
Also to make it clearer, the phrase "(for every grid cell)" has also been added to the following sentence (p5, line 21): "The coefficients of determination ($R^2$) were evaluated for each fitted model (for every grid cell) to identify if a significant level of non-linearity in the input-output relationship was present."

The $R^2$ value is the fraction of the variance of the model output that is explained by the regression model, therefore the closer the $R^2$ value to 1 the better. The choice of the cutoff value for $R^2$ is arbitrary. We would suggest that values of $R^2 > 0.95$ indicate substantial linearity and therefore that regression coefficients of such regression models can be used to link changes in the inputs to the model output response. In the case of our work with the FRAME model, on average there is 2% of variance unexplained by multiple linear regression (4% for $HNO_3$), indicating that non-linearity or interaction terms did not make substantial contributions to the variation in the output for the range of input emissions investigated here.

*(4) What happens to the regression coefficients when the intercept term is not included in the model?*

Response: In the case of perfectly linear response of the FRAME model to changes in the input emissions, multiple linear regression should predict the same model output values for the baseline case as the values produced by the baseline FRAME simulation. If predicted and simulated baseline values are the same then the fractional change relative to the baseline value is 0 for both the inputs and the outputs and the intercept term does not appear in the multiple regression model. For the multiple linear regression models fitted to the data with the input variation ranges corresponding to the uncertainty ranges (± 4%, ± 10% and ± 20% ranges, for $SO_2$, $NO_x$ and $NH_3$) the intercept values were on average 0 (when rounded to two decimal places). Hence, not including the intercept terms would not change the regression coefficients. For the multiple linear regression models fitted to the data with the input variation range of ±40% the intercept values were found to have on average small negative values. This could indicate that some non-linearity in the model response occurs as we move away from the nominal values towards the edges of the input range. However, this non-linearity is not sufficient to make multiple linear regression unsuitable for this analysis.

*(5) In the text line 12 it is stated that the RC 'can be interpreted as the response of an output to a unit change in a particular input when all others are allowed to vary' but in line 25 'RC quantifies the effect of varying a model input X_i alone'. These are contradictory and line 25 is a better description of method 3 (although this section is discussed later).*

Response: We agree that the second description was not correct. As per our response to the related comment (1) above we have changed the second of these sentences to now read (p7, line 7): "RC is a first-order sensitivity measure and it quantifies the average response of model output to varying a model input $X_i$ when all inputs are allowed to vary."

*Method 2 – uncertainty propagation*
*(6) The second method propagates the uncertainty in the emissions to the output using the estimated uncertainties from the UK Informative Inventory Report. Please make it clearer that a new sample has been created here.*

Response: We emphasise that a new LHS sample is created by modifying the text on p5 line 23-24 to: "For the uncertainty propagation, the input sampling space was constrained to the specific uncertainty ranges assigned to the emissions of $SO_2$, $NO_x$ and $NH_3$ in the UK Informative Inventory Report (Misra et al., 2015) with a new LHS sample $n = 100$."

*(7) On line 25 the authors state that the uncertainty is calculated as half the 95% CI relative to the mean value. Half the 95% CI gives 2 \sigma – why is this used as opposed to \sigma? Can't \sigma be calculated directly from the data as I assume it is used to calculate the 95% CI in the first place?*

We followed guidelines for uncertainty reporting as recommended by:

IPCC: IPCC Guidelines for National Greenhouse Gas Inventories, General Guidance and Reporting. [online] Available from: https://www.ipcc-nggip.iges.or.jp/public/2006gl/pdf/1_Volume1/V1_3_Ch3_Uncertainties.pdf, 2006

and

Pulles, T. and Kuenen, J.: EMEP/EEA air pollutant emission inventory guidebook. [online] Available from: https://www.eea.europa.eu/publications/emep-eea-guidebook-2016, 2016.

Either σ, 2σ or a confidence interval is equally suitable for presenting uncertainty where the resulting probability distribution function of the variable of interest is symmetrical (as is the case for FRAME outputs) and we chose the latter. Of course it is always important to specify which is being used, as we have done. (When the PDF is not symmetrical then upper and lower limits of the confidence interval can be specified separately to indicate the resulting uncertainty.)

*(8) It would be helpful to see the emissions maps – even if just in the supplementary material.*

Response: Emission maps for $SO_2$, $NO_x$ and $NH_3$ have been added to the Supplementary Material (Figure S1).

*Method 3 - SRC*
*(9) I need some more convincing that the SRCs calculated here are the same as the measures from Saltelli 2008 – can you expand this? It is these measures that are normally be referred to as the main effects.*

Response: Sections 1.2.5-1.2.8 in the following reference provides the algebraic demonstration that standardised regression coefficients can be equated to first-order sensitivity indices where the model under investigation is linear (as is the case for our work here), i.e. that the following is true.

$$S_{Xi} = \frac{V_{X_i}(E_{X_{\sim i}}(Y|X_i))}{V(Y)} = \beta_{Xi}^2$$

The equation holds for linear models with $S_{Xi}$ being a model-free generalisation. For non-linear models $\beta_{Xi}^2$ (SRC) differs from $S_{Xi}$.

Additionally, it is demonstrated by Borgonovo (2006) that variance-based sensitivity measure coincides with SRC for a linear model with first-order terms.

Saltelli, A., Ratto, M., Andres, T., Campolongo, F., Cariboni, J., Gatelli, D., Saisana, M. and Tarantola, S. (2008) Global Sensitivity Analysis. The Primer, John Wiley & Sons, Ltd, Chichester, UK.

Borgonovo, E.: Measuring uncertainty importance: Investigation and comparison of alternative approaches, Risk Anal., 26(5), 1349–1361, doi:10.1111/j.1539-6924.2006.00806.x, 2006.

*(10)    What is the R^2 for these new regression fits and how were sigma_i and sigma_Y derived? sigma_i is stated as the standard deviation of the input – it should be the standard deviation of the output given uncertainty in the input.*

Response: By definition, standardised regression coefficients (SRCs) are linear regression coefficients multiplied by the ratio of standard deviation of predictor (input) to standard deviation of dependent variable (output) as shown in Eq. 2 in the paper. For the multiple linear regression performed for the sample with "narrower" ranges in inputs (i.e. reflecting the NAEI estimates of uncertainty in each input variable), the $R^2$ values were found to be higher than for the analyses using the ± 40% input variation ranges. The median $R^2$ values were as follows: lowest 0.992 ($HNO_3$) highest 1.000 ($NO_x$), other output variables 0.996-0.999. This indicates that for smaller deviations from the nominal values of emissions the model response is even closer to linear.

*(11)    I would also expect the measures here to follow the regression coefficients more closely given the linearity in the model. This measure is giving different information to the RC and I don't fully understand what that difference is and why the results are different.*

Response: We assume that in this question the reviewer is asking whether values of SRCs squared should follow same spatial pattern as RCs. The distinction between RCs and SRCs is made in the Methods section (p5). RC, as defined in the paper, enables estimation of the response of the model output to a relative change (within ± 40% range) in one or multiple emission inputs. So it can be interpreted as a scaling coefficient applied to the input to get to the output. The sign of RC indicates if the input-output relationship is direct or inverse. The value of RC depends on the units of inputs and outputs. SRC is unit-independent. SRC squared is used to apportion uncertainty and its value depends not only on the model response to the change in the input, but also the magnitude of variation assigned to that input. For example, the input may be assigned a large uncertainty, but if it is not influential, it will not affect output uncertainty. Hence SRCs squared (here the same as first-order sensitivity indices because the model is linear) indicate the extent to which a particular input with a particular uncertainty assigned to it drives uncertainty in the model output.

*(12)    I think the authors should consider using generalised additive modelling here to calculate the main effects following Strong M, Oakley JE, Brennan A. Estimating multiparameter partial Expected Value of Perfect Information from a probabilistic sensitivity analysis sample: a non-parametric regression approach. Medical Decision Making, 2014;34(3):311-26 particularly Eq 6 and 8.*

Response: The abovementioned paper suggests the use of a nonparametric regression (GAM and Gaussian process) approach to estimate the partial expected value of perfect information (EVPI). This is a good suggestion for models that show non-linear and/or non-monotonic trends in the input-output relationships. In general GAM and other non-parametric models can be used as an emulator to estimate the model output at any point in the input space, allowing Monte Carlo estimation of the variance-based sensitivity indices and/or re-estimation of uncertainty in the output when different ranges of uncertainty are assigned to the inputs.

Technically, a linear regression model can be considered as an emulator as well because it enables estimation of model response at any point in the input space. This would allow for example the calculation of first-order sensitivity indices using the approach described by Saltelli et al. (2010). In the case of the FRAME model, multiple linear regression is sufficient because $R^2$ values indicate that for all FRAME output variables the total variation in the output is sufficiently explained by the fitted model. Moreover, SRCs squared can be equated to firstorder sensitivity indices for a linear model, which eliminates the need for further re-sampling and calculations.

To acknowledge the fact that emulators can be used for computationally expensive models with non-linear/non-monotonic response to changes in the inputs, the text at the end of the Methods section has been modified and extended (with additional citations) as follows (p6, line 13-17):
"For the case of non-linear models, variance decomposition methods are described in more detail elsewhere (Homma and Saltelli, 1996; Saltelli, 2002; Saltelli et al., 2010; Sobol', 1993). In the case where a large number of model simulations is not possible an emulator based approach can be used for the uncertainty and sensitivity analysis (Blatman and Sudret, 2010; Lee et al., 2011; Shahsavani and Grimvall, 2011; Storlie and Helton, 2008)."

Saltelli, A., Annoni, P., Azzini, I., Campolongo, F., Ratto, M., & Tarantola, S. (2010). Variance based sensitivity analysis of model output. Design and estimator for the total sensitivity index. Computer Physics Communications, 181(2), 259–270. https://doi.org/10.1016/j.cpc.2009.09.018

Blatman, G. and Sudret, B.: A comparison of three metamodel-based methods for global sensitivity analysis: GP modelling, HDMR and LAR-gPC, Procedia - Soc. Behav. Sci., 2(6), 7613–7614, doi:10.1016/j.sbspro.2010.05.143, 2010.

Lee, L. A., Carslaw, K. S., Pringle, K. J., Mann, G. W. and Spracklen, D. V.: Emulation of a complex global aerosol model to quantify sensitivity to uncertain parameters, Atmos. Chem. Phys., 11(23), 12253–12273, doi:10.5194/acp-11-12253-2011, 2011.

Shahsavani, D. and Grimvall, A.: Variance-based sensitivity analysis of model outputs using surrogate models, Environ. Model. Softw., 26(6), 723–730, doi:10.1016/j.envsoft.2011.01.002, 2011.

Storlie, C. B. and Helton, J. C.: Multiple predictor smoothing methods for sensitivity analysis: Description of techniques, Reliab. Eng. Syst. Saf., 93(1), 28–54, doi:10.1016/j.ress.2006.10.012, 2008.

(13)     *The references should be expanded to include other uses of sensitivity analysis in earth science models. These tools are generally applicable across different types of models which is an important point to make.*

Response: We have added the following text and citations to the Introduction (p 3, line 20-22):
"Global sensitivity and uncertainty analyses have been applied in many earth science fields such as hydrological modelling (Shin et al., 2013; Yatheendradas et al., 2008), ecological modelling (Lagerwall et al., 2014; Makler-Pick et al., 2011; Song et al., 2012), and atmospheric aerosol modelling (Carslaw et al., 2013; Chen et al., 2013; Lee et al., 2011)."

Shin, M. J., Guillaume, J. H. A., Croke, B. F. W. and Jakeman, A. J.: Addressing ten questions about conceptual rainfall-runoff models with global sensitivity analyses in R, J. Hydrol., 503(2013), 135–152, doi:10.1016/j.jhydrol.2013.08.047, 2013.

Yatheendradas, S., Wagener, T., Gupta, H., Unkrich, C., Goodrich, D., Schaffner, M. and Stewart, A.: Understanding uncertainty in distributed flash flood forecasting for semiarid regions, Water Resour. Res., 44(5), doi:10.1029/2007WR005940, 2008.

Lagerwall, G., Kiker, G., Muñoz-Carpena, R. and Wang, N.: Global uncertainty and sensitivity analysis of a spatially distributed ecological model, Ecol. Modell., 275, 22–30, doi:10.1016/j.ecolmodel.2013.12.010, 2014.

Makler-Pick, V., Gal, G., Gorfine, M., Hipsey, M. R. and Carmel, Y.: Sensitivity analysis for complex ecological models – A new approach, Environ. Model. Softw., 26(2), 124–134, doi:10.1016/j.envsoft.2010.06.010, 2011.

Song, X., Bryan, B. A., Paul, K. I. and Zhao, G.: Variance-based sensitivity analysis of a forest growth model, Ecol. Modell., 247, 135–143, doi:10.1016/j.ecolmodel.2012.08.005, 2012.

Carslaw, K. S., Lee, L. A., Reddington, C. L., Pringle, K. J., Rap, A., Forster, P. M., Mann, G. W., Spracklen, D. V., Woodhouse, M. T., Regayre, L. A. and Pierce, J. R.: Large contribution of natural aerosols to uncertainty in indirect forcing, Nature, 503(7474), 67–71, doi:10.1038/nature12674, 2013.

Chen, S., Brune, W. H., Lambe, A. T., Davidovits, P. and Onasch, T. B.: Modeling organic aerosol from the oxidation of $\alpha$-pinene in a Potential Aerosol Mass (PAM) chamber, Atmos. Chem. Phys., 13(9), 5017–5031, doi:10.5194/acp-13-5017-2013, 2013.

Lee, L. A., Carslaw, K. S., Pringle, K. J., Mann, G. W. and Spracklen, D. V.: Emulation of a complex global aerosol model to quantify sensitivity to uncertain parameters, Atmos. Chem. Phys., 11(23), 12253–12273, doi:10.5194/acp-11-12253-2011, 2011.